# Decoding the Nucleolar Role in Meiotic Recombination and Cell Cycle Control: Insights into Cdc14 Function

**DOI:** 10.3390/ijms252312861

**Published:** 2024-11-29

**Authors:** Paula Alonso-Ramos, Jesús A. Carballo

**Affiliations:** 1Centro de Investigaciones Biológicas Margarita Salas, CSIC, 28040 Madrid, Spain; palons03@ucm.es; 2Instituto de Biología Funcional y Genómica, IBFG, CSIC-USAL, 37007 Salamanca, Spain

**Keywords:** Meiosis, Nucleolus, CDKs, Cdc14, Homologous Recombination, Cdc5

## Abstract

The cell cycle, essential for growth, reproduction, and genetic stability, is regulated by a complex network of cyclins, Cyclin-Dependent Kinases (CDKs), phosphatases, and checkpoints that ensure accurate cell division. CDKs and phosphatases are crucial for controlling cell cycle progression, with CDKs promoting it and phosphatases counteracting their activity to maintain balance. The nucleolus, as a biomolecular condensate, plays a key regulatory role by serving as a hub for ribosome biogenesis and the sequestration and release of various cell cycle regulators. This phase separation characteristic of the nucleolus is vital for the specific and timely release of Cdc14, required for most essential functions of phosphatase in the cell cycle. While mitosis distributes chromosomes to daughter cells, meiosis is a specialized division process that produces gametes and introduces genetic diversity. Central to meiosis is meiotic recombination, which enhances genetic diversity by generating crossover and non-crossover products. This process begins with the introduction of double-strand breaks, which are then processed by numerous repair enzymes. Meiotic recombination and progression are regulated by proteins and feedback mechanisms. CDKs and polo-like kinase Cdc5 drive recombination through positive feedback, while phosphatases like Cdc14 are crucial for activating Yen1, a Holliday junction resolvase involved in repairing unresolved recombination intermediates in both mitosis and meiosis. Cdc14 is released from the nucleolus in a regulated manner, especially during the transition between meiosis I and II, where it helps inactivate CDK activity and promote proper chromosome segregation. This review integrates current knowledge, providing a synthesis of these interconnected processes and an overview of the mechanisms governing cell cycle regulation and meiotic recombination.

## 1. Introduction to the Cell Cycle and Its Regulatory Processes

To fulfil their essential functions, organisms require their cells to undergo a series of division and proliferation processes collectively known as the cell cycle. This cycle facilitates the transmission of genetic information, encoded within deoxyribonucleic acid (DNA) molecules, from one generation to the next, ensuring that daughter cells inherit the same genetic material as their parent cell. The cell cycle is organized into four distinct phases: G1, S, G2, and M. During the G1 phase, the cell develops and prepares for DNA synthesis; during the S phase, DNA replication occurs. In the G2 phase, the cell prepares for mitosis—and in mitosis, the replicated DNA is precisely segregated and passed on to descendant cells (Figure 1; [1]).

Mitosis is subdivided into five sub-phases:-Prophase, characterized by the condensation and individualization of chromosomes.-Prometaphase, identified by bipolar spindle assembly and the progressive alignment of chromosomes.-Metaphase, during which chromosomes align along the equatorial plane of the cell, the metaphase plate, and attain their maximum level of condensation.-Anaphase, where chromosomes are equally segregated to opposite poles of the cell. Anaphase is itself subdivided into anaphase A and B.-Telophase is the stage of mitosis in which chromosomes decondense and the nuclear envelope re-forms around each set of chromosomes. The cell begins to prepare for cytokinesis, the process of cytoplasmic division (Figure 1).

The cell cycle is a tightly regulated process that relies on molecular mechanisms promoting cell cycle progression, along with checkpoints that act as quality controls. These checkpoints slow down or halt cell cycle progression if the preceding phase is incomplete or if errors are detected. Progression through the cell cycle is primarily driven by the activity of heterodimeric proteins composed of a regulatory subunit (cyclin) and a catalytic subunit (Cyclin-Dependent Kinase, CDK; [1]). Cell cycle checkpoints function by inhibiting CDK activity [2]. The roles of both checkpoints and cyclin/CDK complexes will be examined in more detail below.

The cell cycle and its regulatory mechanisms are universally conserved in eukaryotes, making them ideal subjects for study in various model organisms, starting with the yeasts *Saccharomyces cerevisiae* and *Schizosaccharomyces pombe*, amphibians such as *Xenopus laevis*, and the fruit fly *Drosophila melanogaster*. In addition to these models, mammalian cell lines such as HeLa and NIH 3T3 cells are extensively used to study cell cycle regulation in higher eukaryotes. Other valuable models include *Caenorhabditis elegans*, *Danio rerio* (zebrafish), and various plant species such as *Arabidopsis thaliana*, *Zea mays*, and *Triticum aestivum*, which provide insights into conserved and divergent cell cycle mechanisms across different biological kingdoms. Among all these model organisms, yeasts are particularly advantageous due to their haploid form, which facilitates the targeted mutation of specific genes, and their ability to be easily cultured in synchronous conditions [1].

## 2. Meiosis

Meiosis is a specialized form of cell division occurring in sexually reproducing organisms, essential for the generation of gametes. This process ensures a reduction in chromosome number by half, a critical step for maintaining genomic stability across generations. Meiosis involves two sequential nuclear divisions—meiosis I and meiosis II—following a single round of DNA replication, ultimately producing four haploid cells from an original diploid progenitor cell [3]. During the first meiotic division, homologous chromosomes segregate toward opposite poles, facilitated by the attachment of sister chromatid kinetochores from each homolog to microtubules originating from the same spindle pole. Conversely, the sister chromatids of the opposing homolog within the same bivalent attach to microtubules from the opposite spindle pole, resulting in reductional chromosome segregation without an intervening round of DNA replication. The second meiotic division is characterized by the separation of sister chromatids, an equational division that mirrors mitosis, further refining the haploid chromosome set [4].

During the meiotic S phase, genetic material is replicated [5]. This S phase is generally longer than in mitosis, allowing for essential interactions between homologous chromosomes. These interactions are crucial for meiotic recombination and the accurate segregation of homologous chromosomes [3,6].

Although meiosis is a continuous process, it can be compartmentalized into distinct phases depending on the different processes that are taking place.

The first meiotic division encompasses prophase I, metaphase I, anaphase I, and telophase I. During this division, homologous chromosomes undergo segregation toward opposite poles.

Prophase I, the most protracted phase of meiosis, is subdivided into several distinct sub-phases. The initial sub-phase, leptotene, is characterized by the cytological appearance of chromosomes as fine threads under the microscope. During leptotene, chromosomes undergo condensation but have not yet paired with their homologous counterparts. This phase is marked by the initiation of genome fragmentation through the introduction of double-strand breaks (DSBs) in the DNA. These DSBs are generated by the enzyme Spo11. The introduction of DSBs is a prerequisite for homologous DNA recombination, facilitating genetic exchange between homologous chromosomes [7,8,9,10].

This is followed by the zygotene phase, during which the formation of the synaptonemal complex (SC) begins. The SC is a proteinaceous structure crucial for homologous chromosome pairing and recombination, and it is composed of several distinct components that will be discussed in detail later in this review. In organisms such as budding yeasts and mice, successful recombination is essential for the proper assembly of the SC. Specifically, in budding yeast, DSBs must occur prior to the formation of the SC. Conversely, in other species like *Drosophila melanogaster* and *Caenorhabditis elegans*, the SC can form independently of recombination events [11,12].

Once the SC is fully established, the cell progresses to the pachytene stage (Figure 2), characterized by the visualization of the bivalents where synapsis is taking place [9,13]. At the conclusion of pachytene, the SC disassembles, and the repair of the DSBs is completed, leading to the formation of crossovers (COs) [3,14]. Following crossover formation, homologous chromosomes undergo further condensation and begin to separate, except at the chiasmata, which represent the physical manifestation of crossover events. This stage is known as diplotene (Figure 2). Subsequently, during diakinesis, the nucleolus and nuclear envelope begin to disassemble. However, this dissolution does not occur in all eukaryotes; for instance, in the yeast *Saccharomyces cerevisiae*, the nuclear envelope remains intact throughout meiosis [3,15,16].

Upon the conclusion of prophase I, the cell transitions into metaphase I (Figure 2), during which the bivalents achieve their maximum level of condensation. In *Saccharomyces cerevisiae*, during metaphase I, sister kinetochores are attached to the same Spindle Pole Body (SPB)—the yeast equivalent of the centrosome in animal cells—thereby preventing their separation during anaphase I [6]. For proper segregation of homologous chromosomes in anaphase I, cohesion between the chromosome arms must be abolished [6]. Additionally, sister kinetochores must be oriented in the same direction, a process involving the Monopolin complex and associated proteins [17,18,19]. Meanwhile, sister chromatids remain tethered at the centromeric region by cohesins until the onset of anaphase I [20]. Finally, during anaphase I to telophase I, the separated chromosomes migrate to opposite poles (Figure 2).

During the first meiotic division, the cohesin complex holds the sister chromatids together by forming a multiprotein ring around both DNA molecules. This complex is formed by the following proteins: Rec8, Smc1, Smc3, and Scc3. The proteolytic degradation of Rec8 allows chromatid separation during chromosome segregation, and this is carried out by the protease Esp1, also known as separase. Esp1 is inactive until anaphase I and anaphase II when its inhibitor, securin Pds1, is degraded by the action of APC/C (Figure 2), thus allowing meiosis to progress. During anaphase I, Rec8 is protected at the centromeres by Sgo1 and PP2A [6], thus allowing the sister chromatids to stay together and migrate to the same pole when separase acts.

The second meiotic division is similar to mitosis. During prophase II, the chromosomes condense, SPBs/centrosomes migrate to opposite poles and duplicate, the meiotic spindle is reassembled, and in most eukaryotic organisms, the nuclear envelope is, again, disorganized. During metaphase II, the chromosomes are oriented at the equatorial plate, and each chromosome attaches to a fiber of the meiotic spindle so that when the cell enters anaphase II, the sister chromatids will separate and migrate to opposite poles of the cell. The second meiotic division ends in telophase II: the achromatic spindle is disorganized, the nuclear envelope and nucleolus are reorganized, except for organisms with closed meiosis such as yeast, and chromosome decondensation occurs, returning the chromosomes to chromatin ([21]; Figure 2). Finally, cytokinesis occurs, and the cell divides its cytoplasm into four daughter cells [21]. In the case of the yeast *S. cerevisiae* after the second meiotic division, cytokinesis does not take place, and instead, the four haploid nuclei are enveloped in a protective spore wall, forming what is known as a spore tetrad. The spore wall is a tough outer layer that provides protection in adverse conditions [16].

## 3. Regulation of Meiosis 

In highly evolved eukaryotic organisms, the process of meiosis is very complex and is integral to the developmental stages of multicellular organisms, specifically in gametogenesis. In simpler eukaryotes, such as yeast, the primary factor that triggers the onset of meiosis is the availability of nutrients in their environment.

Throughout this review, we will focus primarily on the pathways identified in the yeast *Saccharomyces cerevisiae*, the biological system in which many of these pathways were originally discovered. This review will mainly describe processes in *Saccharomyces cerevisiae*, with occasional references to other organisms, noting whether these processes are conserved across species.

### 3.1. Initiation of Meiosis

Although there are exceptions, most eukaryotic organisms are preferentially diploid in their somatic proliferative state. Many yeasts, including *S. cerevisiae*, can exist as both haploid and diploid in the vegetative state, i.e., when proliferating. Haploid cells have two distinct mating types (MAT), a and α [22]. Cells of opposite mating types can fuse, giving rise to a diploid cell, which, under nutrient-restricted conditions, can initiate the meiotic program. These conditions are the absence of glucose and nitrogen sources and the presence of a non-fermentable carbon source [6,22,23]. Under these conditions, the Anaphase Promoting Complex/Cyclosome (APC/C) destroys the Ume6 protein (subunit of the histone deacetylase complex) required for the transition between mitosis and meiosis (Figure 2). This allows *IME2*, a serine/threonine kinase required for meiosis activation, to be transcribed and further stimulates the destruction of Ume6 by APC/C, thus allowing the yeast to enter meiosis [24,25].

In order for meiosis to occur, cells adapt the regulatory mechanisms of mitosis to meiosis [24]. In the case of the budding yeasts, they carry out several of the following strategies:

On the one hand, some of the mitotic proteins are replaced by meiosis-specific proteins such as the cohesin complex subunit Rec8, which replaces Mcd1 or Scc1 [26]. This allows Rec8 to maintain centromere cohesion during the first meiotic division.

Alternatively, mitosis-specific proteins acquire different functions during meiosis, such as the cyclins Clb5 and Clb6, which are required for the initiation of recombination and SC formation [27].

Another example is APC/C which controls the exit from mitosis and in meiosis has a role in the induction of genes involved in the early stages of transcription, such as promoting progression during the first meiotic division [24].

For a more comprehensive overview of the pathways regulating meiotic initiation in budding yeast, the following reviews are recommended for further reading [23,28].

### 3.2. Role of CDK in Regulation of Meiosis

Post-translational modifications, such as phosphorylation and dephosphorylation, mediate a significant portion of cell cycle regulation, independent of gene expression, transcription, and translation. As in mitosis, in meiosis, regulation is achieved through combinations of specific cyclin/CDK pairs. The variation in CDK activity and specificity is determined by cyclins, which are only present and active at the stage of meiosis at which they act [29,30].

In the budding yeast *S. cerevisiae*, there are nine types of cyclins, the G1 cyclins (Cln1, Cln2, and Cln3) and the B-type cyclins (Clb1, Clb2, Clb3, Clb4, Clb5, and Clb6). Neither G1 cyclins nor Clb2 are expressed in meiosis: they are specific to mitosis [30].

For meiosis to progress in *S. cerevisiae*, the appropriate combinations of CDKs and cyclins must be activated, a process that will in turn allow the activation of another serine/threonine kinase, Ime2. Among the most important CDKs is Cdk1, known as Cdc28 in *S. cerevisiae*, which functions in both mitosis and meiosis [29,30].

To initiate meiosis, Ime2 must activate Cdc28, among other targets, enabling its binding to Clb5 and Clb6 to form the CDK-S complex. This complex is crucial for DNA replication and the onset of recombination [31,32]. In prophase I, the expression of late-stage cyclins like *CLB1* and *CLB3* must be suppressed to prevent the premature assembly of the meiotic spindle and early interaction between kinetochores and microtubules [30,33]. After the completion of DNA replication and late in the meiotic recombination process, Clb1 levels must increase, as this cyclin is the main cyclin involved in the first division (Figure 3). Although Clb1 levels remain high throughout most of the meiotic cycle after prophase I (with a decrease during anaphase I), it is only functional during the first division. This suggests that post-translational mechanisms regulate its activity [29,34]. Clb3 is not translated until the second meiotic division, being the main cyclin to act at this stage [30]. Clb4 accumulates during the first meiotic division and is present throughout meiosis. It is mainly active during the first meiotic division, with reduced activity during the second division, implying that its activity is regulated by post-translational mechanisms. With respect to Clb5, Clb5 concentration and Cdk1-Clb5 activity increases during metaphase I and II and decreases during anaphase I and II (Figure 3), its regulation being determined by the concentration of Clb5 itself [29,30].

As previously mentioned, CDK activity is modulated by cyclins, which can be regulated by different mechanisms. They can have post-translational modifications such as phosphorylation that cause the activity of CDK/cyclin complexes to be downregulated during anaphase I. Another mechanism, involving the APC/C, operates through the co-activators Cdc20 and Ama1 to mark cyclins for ubiquitination, targeting them for degradation by the proteasome during anaphase I and anaphase II [30,35]. Cyclin function can also be regulated through the precise timing of their transcript expression. In this case, the transcription factor Ndt80 is responsible for this role in *S. cerevisiae*, which occurs during prophase I, specifically in the pachytene stage [36,37]. Finally, the recombination checkpoint delays cells in prophase I to provide additional time for DSB repair by directly inhibiting CDKs or suppressing cyclin transcription [30,38].

When cyclin degradation occurs and CDK activity decreases, cell cycle phosphatases such as Cdc14 become activated during the first meiotic division. This will reverse the phosphorylation of CDK substrates, facilitating meiotic spindle disassembly during the first division, along with other functions [18,34]. During the second meiotic division, when CDK activity levels drop again, APC/C^Ama1^ is activated, allowing the disassembly of the meiotic spindle and the reactivation of Cdc14. This triggers the exit from meiosis and the initiation of spore wall formation [39].

### 3.3. Control of Prophase I During the First Meiotic Division

The transition between prophase I and metaphase I is a critical stage in meiosis. Errors during this phase can result in the cell entering metaphase I with unrepaired DSBs, which would be harmful to the cell.

In *S. cerevisiae*, the exit from prophase I is controlled by Cdc28 and the transcription factor Ndt80 [40,41], which activates more than two hundred genes, including *CDC5* and the cyclin genes *CLB1*, *CLB3*, *CLB4*, and *CLB5* [37,41].

Mutants carrying deficient or null alleles for *CDC28* and *NDT80*, respectively, are blocked at the end of pachytene, with SPBs duplicated and homologous chromosomes completely synapsed. This is because there is a decrease in CDK activity; however, *ndt80*∆ mutants have no problems in the initiation of the recombination process, although they tend to accumulate a high number of unresolved recombination intermediates (RIs) and a much lower number of COs [42].

Errors in recombination or chromosome synapsis activate the Mec1-mediated checkpoint, leading to hyperphosphorylation and the reduced abundance of Ndt80. This inhibits Ndt80 activity, resulting in a prophase I arrest [43].

As previously mentioned, Ndt80 is a transcription factor responsible for activating genes that encode Cdc5 polo kinase and B-type cyclins in meiosis. In mitotic cells, Cdc5 is crucial for mitotic exit and cytokinesis and is involved in both the FEAR (cdc-Fourteen Early Anaphase Release) and MEN (Mitotic Exit Network) pathways, which will be discussed in more detail later. To facilitate mitotic exit, Cdc5 acts on a kinase within the MEN pathway, promoting the phosphorylation of its substrates. For cytokinesis, Cdc5 facilitates the localization of a GTP-binding protein to the budding neck, and when this process is disrupted, cytokinesis fails to occur [44,45,46]. In addition to its role in mitosis, Cdc5 plays a prominent role in meiosis. Cdc5 is key during prophase I, promoting pachytene exit and CO formation [29,47,48,49].

In an *ndt80*∆ genetic background, the ectopic induction of *CDC5* expression leads to the resolution of RIs into COs and promotes the disassembly of the SC [50,51,52,53,54]. Additionally, *CDC5* can regulate *NDT80* expression itself [55]. The key functions of *CDC5* include the following: (a) promoting kinetochore orientation during metaphase I, (b) removing cohesins from chromosome arms during anaphase I, (c) inducing the resolution of RIs into COs, and (d) facilitating exit from the first meiotic division [50,51,52,53,54].

### 3.4. Control of the Metaphase I-to-Anaphase I Transition

Another important point of control in meiosis is the transition from metaphase I to anaphase I, where the APC/C macrocomplex is involved. APC/C is a ubiquitin ligase that promotes degradation by the proteasome of numerous proteins, including cyclins and securin. It has functions in both mitosis and meiosis [24,56]. APC/C regulation is mediated by phosphorylation processes, as well as activators and inhibitors. In mitosis, APC/C activators include Cdc20 and Cdh1, which bind sequentially to APC/C conferring substrate specificity. When cyclin/CDK activity is present, Cdc20 activates APC/C by promoting the transition between metaphase and anaphase, for which it carries out the destruction of securin (Pds1), leading to the disassembly of the cohesin complex [56].

After anaphase, Cdh1 acts, which binds to APC/C (APC/C^Cdh1^) and promotes the degradation of B-type cyclins and other proteins, allowing the cell to exit mitosis and enter the G1 phase [24,56]. During the S and G2 phases, APC/C is inactive to allow the proteins needed to build up the SPB to accumulate [24]. In other organisms, APC/C activation is regulated by different co-activators: in *Schizosaccharomyces pombe*, the activator is Mfr1, while in *Drosophila melanogaster*, Fzr2 and Cortex serve this role [56,57].

The Spindle Assembly Checkpoint (SAC) plays a crucial role in regulating the APC/C by inhibiting the onset of anaphase. It does this by preventing the degradation of cyclin B and securin until the kinetochores are properly aligned and oriented [24,56,58]. APC/C regulation is further influenced by the phosphorylation of Cdc20 by Protein Kinase A (PKA). In the presence of DNA damage, Cdc20 fails to associate with Clb2, resulting in a blockade of cells at the metaphase stage. This blockade occurs because PKA phosphorylates Cdc20 at serine residues 52 and 88, thereby preventing the APC/C from binding to its substrates [59].

During meiosis, the regulation of APC/C differs from that observed in mitosis due to the absence of DNA synthesis between the first and second meiotic divisions, necessitating specific regulatory mechanisms for APC/C. Additionally, CDK activity must be maintained at low levels during the first meiotic division to facilitate spindle disassembly while simultaneously remaining sufficiently high to inhibit DNA replication [58]. The regulation of APC/C, in addition to Cdc20, in *S. cerevisiae* is provided by meiosis-specific activators such as Ama1, whose ortholog in *Schizosaccharomyces pombe* is Fzr1 and in *Caenorhabditis elegans* is Fzy-1 (Figure 2; [24]).

Cdc20 is essential for the degradation of Pds1 and the activation of separase during both meiotic divisions (Figure 2; [60]). If the release of Pds1/securin by separase does not occur, the meiosis-specific cohesin, Rec8, will not be cleaved from the chromosome arms during the first meiotic division [52,61]. The conditional deletion of *CDC20* in meiosis (using the *pCLB2-CDC20* construct) leads to yeast cells being arrested in metaphase I, accompanied by increased levels of Pds1 [60]. Cdc20 acts in both the first and second meiotic divisions (Figure 2). This implies that its levels should oscillate during meiosis so as to allow Pds1 to accumulate until metaphase I and be destroyed in anaphase I and then act again in the second division [24]. In meiosis, *CDC20* expression is under the control of the transcription factor *NDT80* [24,36].

In contrast, the action of Ama1 is more delayed (Figure 2). For the activation of APC/C^Ama1^ to occur, it is necessary for cyclins to be degraded first by APC/C^Cdc20^. Furthermore, Cdc20 will be targeted for degradation by APC/C^Ama1^ at the conclusion of the second meiotic division [57].

## 4. The Synaptonemal Complex

The synaptonemal complex is a meiosis-specific multiprotein structure that forms between homologous chromosomes during prophase I [62]. It is a tripartite structure consisting of lateral elements (LEs), transverse filaments (TFs), and the central element (CE; Figure 4; [63,64]). The LEs are known as axial elements (AEs) before synapsis takes place. The area between the LEs and the CE is called the central region and is composed of fibrils [3,63]. Most SC proteins are expressed only during meiosis. The structure of the SC is evolutionarily conserved, but there is little homology between the proteins that comprise it in different species [65,66].

The SC is necessary for DSBs to mature into COs at the end of the pachytene, coinciding with SC disassembly. COs supply the physical linkage that holds homologous chromosomes together until they separate at the end of the first meiotic division [5,67]. The SC begins to form from short, individual axial segments at specific points on the chromosomes, specifically in the areas where COs will take place, during zygotene [66,68,69]. These short segments then elongate along the entire length of the chromosomes and form the tripartite structure of the SC, which can be observed in pachytene [5,66,70]. The SC disassembles during diplotene. In diakinesis, chromosomes prepare for the segregation that will take place during anaphase I [66].

LEs comprise a number of proteins located between the axes of homologous chromosomes and are necessary for the correct assembly of the central region of the SC [62]. LEs contain proteins such as Red1 and Hop1 in *S. cerevisiae* [71,72]. The orthologs of Red1 in other organisms are Rec10/Rec27 in *Schizosaccharomyces pombe*, SYCP2/3 in mice, and ASY3/4 in *Arabidopsis thaliana*; however, these proteins have been lost in *Caenorhabditis elegans* and *Drosophila melanogaster* [66,73,74,75]. LEs assemble from cohesin complexes which recruit Red1 and Hop1 to the chromosome axis [5,63]. These scaffold proteins have very little homology with respect to their sequence but share an N-terminal domain, a central disordered region, and a small C-terminal coiled-coil-like domain [66].

The *RED1* gene in *Saccharomyces cerevisiae* encodes a protein with limited homology to other known proteins. Despite this, all *RED1* homologs share a globular N-terminal domain containing an Armadillo repeat and a Pleckstrin homology domain. Additionally, the protein contains closure motifs responsible for recruiting HORMA proteins, such as Hop1, to the chromosome axes. This is followed by a disordered region and a C-terminal domain, which facilitates the assembly of axial-associated proteins [66]. Red1 interacts with the meiotic cohesin complex and forms a homotetramer, which is essential to the assembly of axial filaments. Red1 will be associated with the axial elements of the bivalents throughout the pachytene stage [76,77].

Hop1 is a member of the HORMA domain family of proteins, characterized by an N-terminal domain, a folded central domain, and a flexible C-terminal region. In *Schizosaccharomyces pombe*, the Hop1 analog is also known as Hop1, while in plants, it is referred to as ASY1/PAIR2 [66]. *C. elegans* contain several HORMA domain proteins such as HTP-1, HTP-2, HTP-3, and HIM3, whereas mammals contain two orthologs known as HORMAD1 and HORMAD2. Conformational change in the HORMA domain in these proteins requires the activity of the ATPase Pch2/TRIP13 [66,78,79], which facilitates the localization of Hop1 to the chromosome axes. As synapsis initiates, Hop1 dissociates from the axial elements of chromosomes. This process is mediated, again, by Pch2/TRIP13, which actively disassembles Hop1 [80,81]. The removal of Hop1 from synapsed regions decreases the formation of DSBs, while unsynapsed regions continue to undergo DSB formation driven by the action of Spo11. The loss of Hop1 also leads to the reduced activity of Mek1 kinase, resulting in diminished interhomolog crossing-over formation and promoting the expression of the transcription factor Ndt80, which triggers exit from pachytene and progression through meiosis [66,82,83].

In budding yeast, the formation of the Red1-Hop1 complex is essential for synapsis to occur. Both Red1 and Hop1 physically interact with the kinase Mek1, which is necessary for proper synapsis, directing recombination between homologs and activating the checkpoint. For Mek1 to be activated, Hop1 has to be phosphorylated at threonine 318, a phosphorylation carried out by the yeast homolog of ATR kinase, Mec1 [82,84,85,86,87]. AEs are necessary for chromosome compaction during prophase I and for the correct pairing of homologous chromosomes. They also play a role in the regulation of DSB repair, promoting repair between homologs and preventing sister chromatid exchange [82,83].

The lateral elements are interconnected by TFs. The proteins that form the TFs contain coiled-coil domains, which shorten the distance between the two LEs [62]. Among the transverse filaments, Zip1 in *Saccharomyces cerevisiae* (Figure 4) plays a key role, with orthologs such as SYCP1 in mice, C(3)G in *Drosophila melanogaster*, ZYP1 in *Arabidopsis thaliana*, and SYP-1 and SYP-2 in *Caenorhabditis elegans* all contributing to the core element of the synaptonemal complex [63]. Both SYCP1 and Zip1 form parallel dimers that align along chromosomes. The C-terminal ends of these dimers interact with each other and connect to the lateral elements, while the N-terminal ends contact the central region of the synaptonemal complex (Figure 4; [64,88,89]).

The CE is approximately 100 nm wide, and the oval structures that constitute the recombination nodules can be observed in the center. In yeast, proteins such as Ecm11, Gmc2, or SUMO are found in the CE (Figure 4; [64,90,91]. For proper SC assembly to occur, it is important that the Ecm11 protein is post-translationally modified by SUMOylation. In this way, the SUMO polymeric chains added to Emc11 allow Zip1 to assemble, providing stability to the SC [64].

## 5. The Nucleolus

Within the nucleus, the best-differentiated structure is the nucleolus. This compartment lacks an outer membrane covering it. It consists of an aggregation of macromolecules, where certain proteins bind to rDNA and form a discrete, more or less stable structure, where the processes and interactions characteristic of the nucleolus take place. In addition, it is involved in stress response, genome integrity and stability, and RNP biogenesis [92]. The nucleolus is a structure with a dual nature; since there are moments in which it can have a more static behavior and others in which it is more dynamic, the components that form the nucleolus enter and leave, being in a dynamic equilibrium with the nucleoplasm [93].

The nucleolus is organized around the Nucleolus Organizing Region (NOR; [94]. This is a region containing the sequences of genes coding for rRNAs. Numerous copies of these gene formations are distributed in tandem so that they are highly repeated along the NOR. Each species has its NORs located on different chromosomes. In the case of humans, they are located on the five acrocentric chromosomes 13, 14, 15, 21, and 22 [93,95], while in *S. cerevisiae*, they are located on chromosome XII and contain between 100 and 200 tandemly repeated rDNA sequences [96,97].

The nucleolus is the site where rRNAs are processed and where they are assembled to form ribosome subunits. Within the nucleolus reside proteins involved in rRNA synthesis and transient proteins, which may be inside or outside the nucleolus depending on the time in the cell cycle at which the cell is located. The different localization of these proteins implies different functions [93].

The expression of rDNA genes is associated with the fibrillar regions of the nucleolus, which are considered to be the areas where polymerase I-dependent transcription initiation takes place [93].

### 5.1. Spatial Organization of the Nucleolus

The organization of the nucleolus has been extensively studied. In *S. cerevisiae*, the nucleolus has two compartments (bipartite structure), occupies one-third of the size of the nucleus, and is composed of fibro-granular-like material, but its components are not easily differentiated from each other (Figure 5A; [96,98,99]).

Higher eukaryotes such as mammals have three compartments (tripartite structure), as observed by electron microscopy (Figure 5B; [93,98,100]).

### 5.2. Organization of the Nucleolus in Interphase

The mammalian nucleolus exhibits a tripartite organization. At its core lie the fibrillar centers (FCs), which are predominantly encased by the dense fibrillar component (DFC; Figure 5B). The DFC is characterized by densely packed fibers and may occupy a substantial portion of the nucleolus. Within the DFC, RNA synthesis is thought to occur, as this region houses the active genes, forming structures often referred to as “Christmas trees” [98,99,100]. The third structural component is the granular component (GC; Figure 5B), which constitutes the outermost region and occupies the largest volume of the nucleolus. Multiple copies of the FC and DFC are dispersed throughout the GC (Figure 5B; [93,98]). In addition to these three primary components, chromatin fibers extend into the nucleolus, making contact with the FCs [93]. These nucleolar compartments are encircled by a ring of condensed chromatin, known as the perinucleolar chromatin ring (Figure 5B; [92,93,98]). The structure of the nucleolus varies depending on the stage of the cell cycle and the specific functions being carried out at that time [93].

### 5.3. Nucleolus Dynamics in Mitosis and Meiosis

During the mitosis of higher eukaryotes, the nucleolus undergoes disassembly and reassembly. Nucleolar disassembly is driven by rDNA repression in prophase [93]. This rDNA silencing is regulated by the CDK/cyclin system, specifically by Cdk1/cyclin B kinase [101]. At the end of telophase, the ribosome assembly machinery is reactivated, Cdc14 reverses CDK/cyclin phosphorylation, and nucleologenesis takes place [93,102]. In contrast, in budding yeast (*S. cerevisiae*), the nucleolus does not disassemble. Instead, *S. cerevisiae* has evolved a mechanism involving the Cdc14 phosphatase and the FEAR and MEN pathways to prevent nucleolar disassembly while still allowing its segregation [103].

During meiosis, the behavior of the nucleolus differs from its role in mitosis. In mammals, it has been observed that at the onset of the first meiotic prophase, the nucleolus fragments. As the cell advances through prophase I, these fragments fuse again, driven by chromosome movement [104]. In budding yeast, during anaphase I, the rDNA of the nucleolus stretches between the two opposing poles until complete separation occurs. If separation does not occur during anaphase I, the rDNA and NORs separate together in anaphase II, where the sister NORs are the last regions of the chromosomes to separate [105].

### 5.4. Models of Nucleolus Physical Organization

The maintenance of membrane-less structures, such as the nucleolus, has been the focus of extensive research in recent years [92,93,96,98,106,107,108,109,110,111,112,113]. Various mechanisms have been proposed to explain how these structures maintain their distinctiveness from the surrounding cellular components.

One model proposed to explain the maintenance of membrane-less organelles is the binding model (Figure 6). This model suggests that subcellular compartments lacking membranes are formed through binding and repulsive interactions between proteins and chromatin [96]. A more complex model, which could incorporate the binding model, posits that proteins create bridges between distant chromatin loci or even between different chromosomes, generating loops and leading to polymer-polymer phase separation (PPPS) [92,96,98]. Another prominent model is liquid–liquid phase separation (LLPS), where membrane-less compartments form through phase separation, creating liquid-like droplets (Figure 6).

In the LLPS model, proteins are not bound to chromatin [96,114] but may be excluded from the nucleolus due to their biochemical properties. In contrast, in the PPPS and binding models, proteins exclude each other primarily based on size. This suggests that proteins operating under these models possess charged domains [96,98]. In organelles with LLPS-type organization, the presence of proteins with *Intrinsically Disordered Regions* (IDRs) is critical, as these regions facilitate protein–protein repulsion. IDRs are composed of amino acids lacking a conserved three-dimensional structure [92,107,115].

The formation of the nucleolus may be explained as a combination of the PPPS and LLPS models, where protein–chromatin-binding regions follow a PPPS-type mechanism, while other regions exhibit an LLPS-type separation [96,109,116].

### 5.5. Nucleolus Components

A significant portion of the proteins that form the nucleolus structure are characterized by being flanked by IDRs. The key proteins maintaining nucleolar structure in mammals include fibrillarin, nucleolin, GAR1, and nucleophosmin. Nucleolin, located in the GC, has three distinct domains: an N-terminal domain, a central domain with two RNA-binding regions, and a C-terminal domain rich in arginine and glycine. Its ortholog in *S. cerevisiae* is Nrs1, which plays a role in pre-rRNA processing [99,117,118,119].

Fibrillarin, found in the DFC, has an ortholog in *S. cerevisiae* called Nop1 [120]. Fibrillarin is responsible for rRNA methylation and contains an IDR region at its N-terminal end, rich in glycine and arginine. It also features a GAR (Glycine–Arginine-Rich) domain, which targets the protein to the nucleolus for ribosome synthesis [92,98,99].

Mammalian nucleophosmin (NPM1 or B23) is primarily localized to the GCs and may also be present in the DFCs. Its structure consists of an oligomerized N-terminal region that undergoes pentamerization, along with two IDRs located between the oligomerization domain and the C-terminal end. These IDRs can bind to basic proteins enriched in arginine [98,99]. Nucleophosmin exhibits DNA- and RNA-binding activities and is involved in various cellular processes, including centrosome duplication, genome stability, stress response, apoptosis, and cancer. Its activity can be regulated by phosphorylation [121]. Notably, there appears to be no homolog in budding yeast.

Nucleophosmin and fibrillarin are immiscible with one another, creating two distinct liquid phases that do not mix, similar to the separation of water and oil, which supports the organization of the DFCs and GC [98].

Lastly, the GAR1 protein, initially identified in *S. cerevisiae*, is localized to the DFCs and serves as the primary catalytic unit of ribonucleoprotein complexes involved in the modification of rRNA residues. The specific function of its C-terminal IDR remains unclear [99].

The nucleolus is a dynamic structure that changes throughout the cell cycle. During interphase, nucleolar proteins exhibit continuous flow in and out of the nucleolus. Upon entering mitosis, there is a redistribution of nucleolar proteins [98]. At the conclusion of mitosis, nucleolar proteins transition from prenucleolar bodies to the nucleolus organizing regions, which serve as nucleation sites. rRNA plays a crucial role in nucleolus formation, while small nucleolar RNAs (sno-rRNAs) are important for regulating its homeostasis [98].

### 5.6. Functions of the Nucleolus

The primary function of the nucleolus is the synthesis of ribosomal RNA (rRNA) and the assembly of ribosomes [93,122,123]. Within the nucleolus, rRNA genes and their transcripts coexist alongside proteins involved in rRNA synthesis and ribosome assembly, both within and outside the nucleolar structure [124].

As previously mentioned, the nucleolus is organized around NORs, which consist of tandem genes that code for rRNA. In most species, each NOR gene contains three coding sequences, 18S, 5.8S, and 28S in mammals and 25S in plants, along with internal and external transcription regions and non-transcribed intergenic regions [125]. The transcription of these pre-rRNA subunits is facilitated by RNA polymerase I, with rRNA primarily localized in the fibrillar regions of the nucleolus, where polymerase I initiates transcription.

In addition to rDNA transcription and ribosome assembly, the nucleolus performs various other functions. These include the inactivation of specific chromatin domains, the spatial reorganization of certain genes for recombination, and the regulation of protein movement between the nucleolus and nucleoplasm. The nucleolus also plays roles in DNA repair, RNA transcription by polymerase II, telomere maintenance, stress responses, and apoptosis [124].

Furthermore, the nucleolus significantly contributes to DNA repair by housing DNA-binding proteins that interact with nucleoplasmic DNA repair processes. For instance, the DNA damage response factor NBS1 can enter the nucleolus to inhibit rDNA transcription, while other proteins exit the nucleolus to bind at DSB repair sites [124,126]. Another critical function of the nucleolus pertains to telomeres. Telomerase, a ribonucleoprotein complex associated with telomeres, assembles within the nucleolus and remains there until DSBs are repaired [124].

During cellular stress, such as DNA damage or fluctuations in temperature or acidity, the exchange of proteins between the nucleolus and the rest of the cell can be significantly altered [124,127].

## 6. Meiotic Recombination Process

Recombination is a programmed process that culminates in the formation of COs [14]. These COs are crucial for the correct segregation of chromosomes during the first meiotic division and contribute to increased genetic variability. To facilitate chiasma formation, recombination in meiosis occurs between homologous chromosomes rather than between sister chromatids, which preferentially occurs in mitosis [9,70,82,128,129].

The initiation of meiotic recombination happens through the generation of DSBs mediated by the transesterase Spo11, which functions similarly to topoisomerase VI, a type II topoisomerase identified in archaea [10,130,131]. Spo11 is part of a large heterocomplex, with a catalytic complex comprising three proteins: Rec102, Ski8, and Spo11 itself. This catalytic complex operates as a dimer, with each Spo11 monomer cleaving a DNA strand from both 5′ ends flanking the DSB via a transesterification reaction. This process results in the formation of a phosphodiester bond between the tyrosine in the active site of Spo11 and the 5′ end of the DSB.

The regulation of DSB formation is governed by a dual mechanism: a positive feedforward mechanism, involving CDKs that promote DSB formation, and a negative feedback mechanism, in which Tel1/ATM kinases inhibit DSB formation [132,133,134,135,136,137,138,139,140,141]. These DSBs typically arise at the onset of prophase I (leptotene) and, in most organisms, precede the assembly of the SC [9,70].

In certain cases, such as in female *Drosophila melanogaster* and *Caenorhabditis elegans*, DSB formation and SC assembly are independent processes. In *Drosophila melanogaster*, DSBs are generated after SC formation, while in *Caenorhabditis elegans*, synapsis is not a prerequisite for DSB formation [11,142,143,144].

In all other organisms, chromosome pairing and synapsis require that the process of strand invasion onto the homologous chromosome during recombination has previously been initiated, and sufficient DSBs must be produced for this to take place and for reliable pairing between homologs to occur [145].

The formation of DSBs is influenced by chromosome morphology and is regulated both temporally and spatially; the specific sites where DSBs occur are referred to as hotspots [9]. Following DSB formation by the Spo11 complex, the MRX complex—comprising the proteins Mre11, Rad50, and Xrs2—along with Sae2, cleaves Spo11 along with a short oligonucleotide fragment (Figure 7). Subsequently, the exonuclease Exo1 processes the 5′ ends, leaving the 3′ end protruding. This facilitates the binding of Replication Protein A (RPA) to the single-stranded DNA, which is essential for subsequent processing by Rad51 and Dmc1.

Rad51 and Dmc1 are both RecA-type recombinase proteins, with Rad51 being present in both mitotic and meiotic processes, while Dmc1 is specific to meiosis. Both proteins play crucial roles in homologous chromosome searching and complementary strand invasion [146,147,148,149]. Additionally, the Mei5-Sae3 and Hop2-Mnd1 complexes are involved in this process. Proteins associated with the axial elements, such as Red1, Hop1, Mek1, and cohesins like Rec8, also facilitate recombination by maintaining homolog association during prophase I [148,150].

In the absence of Dmc1, there is an accumulation of unrepaired DSBs because mutants lacking Dmc1 fail to facilitate the invasion of the single-stranded DNA from the 3′ protruding end onto the double-stranded template of the intact homologous chromosome. Consequently, these mutants exhibit defects in strand invasion and the formation of early recombination intermediates [146,151,152]. For Dmc1 to function effectively, the Mei5-Sae3 and Hop2-Mnd1 complexes must be active. The Mei5-Sae3 complex is meiosis-specific and colocalizes with Dmc1, with its ortholog in *Schizosaccharomyces* pombe being Swi5-Sfr1 [153,154,155]. Mutations in either complex result in phenotypes that are similar to, or even more severe than, those observed in *dmc1* mutants, where cells are arrested in prophase I and are unable to repair DSBs [153,154].

The role of Rad51 in meiosis appears to be more structural, as it promotes the binding of Dmc1 to recombination sites and regulates the exchange between DNA strands [147,151,156]. Once Rad51 and Dmc1 are loaded, a DNA–protein nucleofilament is formed that searches for and pairs with homologous sequences in homologous chromosomes to repair DSBs, creating repair intermediates known as D-loops (displacement loops). These D-loops can be processed, leading to the formation of non-crossovers (NCOs) via the synthesis-dependent strand annealing (SDSA) repair pathway, which involves the RecQ family of DNA helicases, Sgs1 and Srs2 (Figure 7).

Alternatively, the D-loop can be processed after the extension of the invading end through DNA synthesis, resulting in an intermediate known as single-end invasion (SEI). Following the capture of the second end of the DSB, this process continues with DNA synthesis and ligation to form the double Holliday Junction (dHJ; Figure 7; [146,152,157]). After dHJ formation, CO events occur, promoting DNA exchange between homologous chromosomes, although some NCO formation may also take place, depending on the orientation in which the dHJ-resolving enzyme cleaves at the junctional intersections (Figure 7; [151]).

In *Saccharomyces cerevisiae*, the resolution pathways for DSBs into CO or NCO products are regulated by the transcription factor Ndt80. Ndt80 is critical for the transition from late prophase I (pachytene) to metaphase I; mutants lacking Ndt80 are arrested in pachytene, leading to the accumulation of dHJs and a reduction in CO levels [41,42,158]. Although Ndt80 activates a broad array of genes, the mere activation of the Cdc5 polo kinase is sufficient to induce the resolution of dHJs into COs, even in the absence of complete meiosis [50,51,159].

In budding yeast, COs are resolved through various complexes, including Mlh1-Mlh3 (MutLγ) in conjunction with PCNA, Exo1, and Sgs1, as well as structure-selective endonucleases (SSEs) such as Mus81-Mms4, Slx1-Slx4, and Yen1 [160,161,162,163,164,165,166]. The accurate resolution of recombination products necessitates the coordinated action of helicases, topoisomerases, and endonucleases [50].

While the mechanisms governing the elimination of various types of recombination intermediates resulting from errors in the DNA repair process are not fully understood in *S. cerevisiae*, it is known that the resolution activity of Mus81-Mms4 requires hyperactivation through Cdc5-mediated phosphorylation following the exit from pachytene [167]. Conversely, the Holliday junction resolvase Yen1 is typically inhibited during prophase I due to phosphorylation by the Cdk1 complex associated with S-phase cyclins, also referred to as CDK-S. Both Mus81-Mms4 and Yen1 exhibit redundant roles in DNA damage repair; however, they operate at distinct stages due to their regulatory mechanisms [168,169].

During both mitosis and meiosis, the activation of Yen1 in anaphase is facilitated by the dephosphorylation of consensus sites for CDK present in its sequence. This dephosphorylation enables the repair of recombination intermediates and ensures accurate chromosome segregation [170].

In mitotic cells of *Saccharomyces cerevisiae*, the subcellular localization of Yen1 is similarly regulated by the same phosphorylation events, a mechanism that is also observed during meiosis. Specifically, CDK-S induces the delocalization of Yen1 from the nucleus during the S phase and most of prophase I. Following the dephosphorylation of these sites, Yen1 can re-enter the nucleus, become activated, and resolve recombination intermediates [50,167].

The dephosphorylation of Yen1 is mediated by the phosphatase Cdc14 [50,171,172,173]. Mutants deficient in Cdc14 exhibit impairments in the separation of homologous chromosomes and sister chromatids. Notably, in the absence of the Mus81-Mms4- and Sgs1-dependent DSB repair pathways, the loss of either Cdc14 or Yen1 results in similar repair defects. This finding suggests that both proteins participate in the same repair pathway during the first meiotic division, thereby establishing a direct role for Cdc14 phosphatase in the recombination process during meiosis [50].

## 7. The Cdc14 Phosphatase

Cdc14 is a dual-specificity phosphatase that dephosphorylates serine and threonine residues and functions in both mitotic and meiotic processes [174]. The gene encoding Cdc14 was initially described by Culotti and Hartwell in 1971 [175] and later identified in a screening study by Hartwell and colleagues in 1974, which revealed a series of genes that regulate cell division within the cell cycle, collectively named the CDC (Cell Division Cycle) [176]. However, it was not until 1997 that Taylor and collaborators elucidated the enzymatic activity of these proteins [177].

The budding yeast Cdc14 is composed of 551 amino acids, including an N-terminal domain and a C-terminal domain. The N-terminal region is conserved across various species and is subdivided into two essential domains for phosphatase activity: the A domain, which ensures substrate specificity, and the B domain, which serves as the catalytic domain. In contrast, the C-terminal region is less conserved and contains a nuclear export signal (NES) and a nuclear localization signal (NLS) (Figure 8; [178,179,180]). Additionally, Cdc14 has a SUMOylation site at lysine 203, and 29 phosphorylation residues have been identified so far [181,182,183].

In *Saccharomyces cerevisiae*, Cdc14 predominantly resides in the nucleolus throughout most of the cell cycle, where it remains inactive and bound to its inhibitor, Cfi1/Net1. It is a component of the Regulator of Nucleolar Silencing and Telophase (RENT) complex [184,185]. The regulation of Cdc14 activity is primarily dictated by its subcellular localization, as its concentration does not fluctuate during the cell cycle [185].

### 7.1. Role of Cdc14 in Mitosis

Cdc14 is essential for cells to successfully terminate mitosis and progress through the cell cycle by reversing the phosphorylation of mitotic CDKs until mitotic spindle elongation is completed [186]. To effectively fulfil this role, Cdc14 must be tightly regulated to prevent premature exit from mitosis [185,187].

To facilitate the transition out of mitosis, Cdc14 dephosphorylates Cdh1, which promotes the activation of the APC/C within the nucleus. Additionally, Cdc14 enhances the accumulation of Sic1, a CDK inhibitor, by dephosphorylating both Sic1 itself and its transcription factor Swi5. This process subsequently promotes the expression of *SIC1* (Figure 9). The combined effects of APC/C activation and increased Sic1 levels lead to the downregulation of CDK activity, ultimately enabling the cell to exit mitosis (Figure 9; [186,188,189,190,191]).

In addition to its role in promoting mitotic exit, Cdc14 is involved in several critical processes during mitosis, including SPB duplication, autophagy, DNA replication, and the regulation of mitotic recombination. Furthermore, it contributes to ribosomal DNA (rDNA) condensation and segregation [191,192,193,194,195].

The regulation of Cdc14 binding to its inhibitor is modulated by phosphorylation events. Specifically, the release of Cdc14 is accompanied by an increase in phosphorylated residues in both Cdc14 and its inhibitor Cfi1/Net1. Notably, Cfi1/Net1 contains six critical phosphorylation sites and their mutation prevents the release of Cdc14 from the RENT complex. The phosphorylation of these sites is mediated by the Cdc5 polo kinase, which phosphorylates Net1/Cfi1 at threonine 212, as well as by components of the FEAR pathway, such as Dbf2-Mob1, which will be discussed later (see below; [188,196,197,198]). These phosphorylation sites consist of S/T-P motifs that align with the minimal consensus sites recognized by CDKs [196]. Upon its release from Cfi1/Net1, Cdc14 translocates to the nucleus and cytoplasm, where it engages with its substrates [185,196,199].

### 7.2. Regulation of Cdc14 by the FEAR and MEN Pathways

The regulation of Cdc14 release from the nucleolus, leading to the phosphorylation of both Cdc14 and Net1, occurs through two primary pathways, the FEAR pathway and the MEN pathway, the latter of which is analogous to the Hippo pathway in higher eukaryotes [200].

During mitosis, the FEAR pathway acts first among the two pathways that regulate Cdc14. This pathway is responsible for the initial transient release of Cdc14 from the nucleolus, occurring in early anaphase [185,199,201].

The key proteins involved in the FEAR pathway include separase Esp1, the kinetochore-binding protein Slk19, Spo12, and the replication fork protein Fob1, which enhances the stability of the interaction between Cdc14 and Cfi1/Net1. Additionally, the FEAR pathway encompasses PP2A, a phosphatase associated with its regulatory subunit Cdc55, as well as Zds1, Zds2, the CDK complexes with cyclins Clb1 and Clb2, and Cdc5 [18,103,187].

The activation of the FEAR pathway is initiated when the APC/C degrades securin (Pds1), which normally inhibits separase (Esp1; Figure 10). The activation of Esp1 is critical for the early release of Cdc14, as it promotes the CDK-mediated phosphorylation of Net1/Cfi1 [199,202]. This subsequently leads to the activation of the Esp1/Slk19 complex. This complex regulates PP2A^Cdc55^ by phosphorylating its regulatory subunit, Cdc55, with the involvement of Zds1 and Zds2 [203,204]. This regulation permits Clb1-CDK and Clb2-CDK complexes to phosphorylate Cfi1/Net1, facilitating the release of Cdc14 from the nucleolus during early anaphase [187,196,203,205].

In parallel, the nucleolar phosphoprotein Spo12 binds to the FEAR pathway inhibitor, Fob1. Fob1 associates with Cfi1/Net1 to inhibit Cdc14, but Spo12 induces a conformational change in Fob1 that allows the release of Cdc14 (Figure 10). This mechanism is driven by the phosphorylation of Spo12 at Serine 118, a modification that is restricted to anaphase and requires the activity of Esp1 and Slk19 [103,186,206].

Thus, Esp1, Slk19, and Spo12 act as positive regulators of the FEAR pathway, along with the polo-like kinase Cdc5. In contrast, Pds1 and the nucleolar protein Fob1 function as negative regulators [103,199].

Importantly, the initial release of Cdc14 appears to be independent of CDK activity levels [199]. Although the release of Cdc14 via the FEAR pathway is not essential for cell survival, mutations in FEAR pathway components significantly compromise cell viability, leading to increased cell death compared to wild-type cells [103,187].

The activation of Cdc14 during late anaphase is governed by the GTPase-driven signaling cascade of the MEN pathway [185]. This pathway is named after the fact that mutations in its components lead to permanent arrest in anaphase, characterized by sustained high CDK activity, preventing cells from completing mitosis [189,207,208].

The MEN pathway comprises several key components: Tem1, a small GTPase; the Bub2-Bfa1 complex; Lte1; two kinases, Cdc15 and Dbf2; and Mob1, a cofactor for Dbf2 (Figure 10). Nud1, a scaffolding protein, ensures the structural integrity of the pathway. The regulation of these components during mitosis is tightly linked to the subcellular positioning of the SPBs, with MEN activation occurring as SPBs migrate into the bud of the dividing yeast cell [103,187,205].

Tem1 localizes to the SPBs, where it initiates MEN activation. The Bub2-Bfa1 complex inhibits Tem1 until the Bfa1 subunit is phosphorylated by Cdc5, enabling Tem1 activation [45,209]. Once activated, Tem1 signals to Cdc15, which in turn activates the Mob1/Dbf2 complex. This cascade results in the second release of Cdc14 from its inhibitor Cfi1/Net1, allowing Cdc14 to translocate into the nucleus and cytoplasm, where it dephosphorylates various CDK substrates to promote mitotic exit (Figure 10; [210]).

Efficient MEN activation requires the prior release of Cdc14 mediated by the FEAR pathway [211]. PP2A^Cdc55^, a component of the FEAR pathway, plays a critical role in dephosphorylating Mob1, a MEN component. Additionally, the downregulation of PP2A^Cdc55^ enhances Bfa1 phosphorylation and its asymmetric localization at SPBs, contributing to MEN pathway activation and progression through mitosis. Notably, a physical interaction between Bfa1 and Cdc55 suggests that when PP2A^Cdc55^ is downregulated by separase, it triggers Cdc14 release via FEAR and MEN activation [211].

Cdc5 functions as a key regulator in both the FEAR and MEN pathways. It has been demonstrated that Cdc5 can act on both Cdc14 and its inhibitor, Cfi1/Net1. Specifically, Cdc5 directly phosphorylates Net1, inhibiting its function during both early and late mitosis. While the exact mechanism of the influence of Cdc5 on Cdc14 remains less defined, it is known that the two proteins physically interact. Moreover, the phospho-serine/phospho-threonine-binding domain of Cdc5 is sufficient to mediate this interaction, suggesting a regulatory role of Cdc5 on Cdc14 [189,197,198,199,212,213].

For mitotic exit to proceed, Cdc5 must phosphorylate and inactivate Bfa1. This inactivation is critical, as it releases the inhibition of Tem1, thus allowing MEN pathway activation. If Bfa1 is not inactivated, cells fail to exit mitosis due to the continued suppression of the MEN pathway [45,197,198,209,212,214].

The transition between the FEAR and MEN pathways is less well understood, but some evidence suggests that Tof2 may play a pivotal role in this process (Figure 10). In addition to its association with Cfi1/Net1, Cdc14 also binds to Tof2. While Tof2 promotes Cdc14 activity, its absence does not completely prevent the release of Cdc14 from the nucleolus [215]. The role of Tof2 appears to be facilitating rDNA segregation by retaining a portion of the active Cdc14 within the nucleolus. This mechanism likely ensures a two-step release of Cdc14, with Tof2 modulating the transition between FEAR and MEN pathway activities, thereby preventing a complete release of Cdc14 during the first wave [215].

Furthermore, Tof2 is essential for proper nucleolar segregation. In cells lacking Tof2, nucleolar division is disrupted, leading to the abnormal distribution of the nucleolus between segregating chromosomes during anaphase and telophase. This defect causes a delay in nucleolar segregation during mitosis, further highlighting the importance of Tof2 in coordinating this process [215,216,217].

### 7.3. Functions of Cdc14 in Meiosis

Similar to mitosis, Cdc14 is sequestered in the nucleolus as part of the RENT complex, where it remains inactive during meiosis. Upon entry into anaphase I, Cdc14 is released into the nucleus, but it returns to the nucleolus by the end of telophase I. It remains there until anaphase II, when it is again released, this time into both the nucleus and cytoplasm [18,218]. Cdc14 is inhibited by its association with Net1/Cfi1, similar to the regulation observed in mitosis. As discussed previously, there are six key CDK phosphorylation sites required for Cdc14 release during mitosis [196,197,198]. Recent research into these phosphorylation sites in meiosis indicates that their mutation impairs Cdc14 release during anaphase I, leading to defects in rDNA segregation when compared to wild-type cells. However, in both wild-type and *net1-6cdk* mutant cells, Cdc14 is still released in late anaphase II, enabling the completion of meiosis, though *net1-6cdk* mutants exhibit a slightly higher number of dyads than wild-type cells [219].

The regulation of Cdc14 during meiosis primarily depends on the FEAR pathway proteins Slk19, Spo12, and Esp1 (Figure 11) [18,34]. Unlike mitosis, where the MEN pathway is essential for maintaining Cdc14 release during anaphase, the FEAR pathway is crucial in meiosis, specifically for the transition between the first and second meiotic divisions [18,22,34,190]. The MEN pathway, however, becomes essential after the second meiotic division, as the absence of key MEN components results in an inability to complete sporulation [190,218,220].

In meiosis, Cdc14 and the FEAR pathway play roles analogous to those in mitosis, facilitating the entry into anaphase I. However, recent studies suggest that their role in reducing Clb-CDK levels may be less critical in meiosis than in mitosis [221]. Nonetheless, the absence of both FEAR pathway components and Cdc14 leads to an inability to lower Clb-CDK levels, which results in defective spindle disassembly, improper chromosome segregation, and impaired nucleolar division [18,34,187,199]. Conversely, the premature activation of the FEAR pathway causes defects in SPB assembly [222,223].

During mitosis, Cdc14 is known to play a role in SPB disassembly [186]. In meiosis, similar findings have been observed. In cells lacking the MEN pathway component Cdc55, Cdc14 is prematurely released from the nucleolus, causing delays in spindle disassembly during the first meiotic division and resulting in minimal spindle disassembly during the second division [221]. These observations suggest that Cdc14 also contributes to the regulation of SPB disassembly during meiosis.

During both anaphase I and anaphase II, Cdc14 exhibits an asymmetric association with the SPBs. In anaphase I, Cdc14 colocalizes with one of the SPBs, while in anaphase II, it is associated with two of the four SPBs. This SPB binding is regulated by the MEN pathway component Bub2-Bfa1 [222]. The association of Cdc14 with SPBs is crucial for the proper transition between the first and second meiotic divisions and for the re-duplication of SPBs [221,222].

As well as in mitosis, Cdc14 plays a role in autophagy, although it is a more specialized role, with a focus on the selective degradation of organelles to ensure gamete quality [224].

### 7.4. Orthologs of Cdc14

Orthologs of Cdc14 have been found in other eukaryotes. In general, the function of downregulating CDK activity is conserved, as is their role in spindle stability and assembly, although they also carry out other functions [180,187].

In *Schizosaccharomyces pombe*, the ortholog of Cdc14 is called Clp1 or Flp1. Like Cdc14 of *S. cerevisiae*, it localizes to the nucleolus, but Clp1/Flp1 is released in early mitosis, and no homologs of the FEAR pathway participate. Its role is not important for mitosis exit but for entry. It also coordinates cytokinesis and the initiation of a new cell cycle [180]. In *Caenorhabditis elegans*, the ortholog is called CeCdc14 (*cdc-14*). In Xenopus, two have been described, XCdc14A and XCdc14B; in birds, there are also two orthologs, cCdc14A and cCdc14B [180].

With respect to mammals, two copies, CDC14A and CDC14B, and a third copy, CDC14C, exclusive to hominids, have been described [225,226]. In humans, they are named hCdc14A, hCdc14B, and hCdc14C. hCdc14A is found in the cytoplasm and, depending on the stage of the cell, may also be in the centrosomes. hCdc14B is located in the nucleolus and centromeres [180].

Cdc14, therefore, is a phosphatase that is widely conserved throughout evolution but whose functions appear to be less conserved.

### 7.5. Roles of Cdc14 in Meiotic Recombination

The timing of joint molecule (JM) resolution and crossover formation during meiosis in budding yeast is tightly coupled with cell cycle progression and regulated by the *NDT80*-dependent activation of the polo-like kinase Cdc5 [51,159]. During the transition from pachytene to metaphase I, Cdc5, activated by Ndt80, not only promotes the resolution of double Holliday junctions but also appears to modulate the interaction between Cdc14 and Cif1/Net1 in the nucleolus. Unlike its role in mitosis, Cdc5 may transiently counteract the inhibitory effect of PP2A^Cdc55^, allowing a fraction of Cdc14 molecules to be released from sequestration [222,223]. This indicates that Cdc5 may induce an early, metastable, partial release of Cdc14 during the pachytene-to-metaphase I transition, thereby fine-tuning the function of enzymes essential for proper DNA repair and chromosome segregation [155,169,227].

A recent study from our lab indicates that depleting Cdc14 during meiosis leads to severe defects in the resolution of recombination intermediates, resulting in a significant decrease in CO formation when alternative repair pathways are compromised. In metaphase I-arrested *cdc20-md* cells, Cdc14 is essential for proper RI resolution even prior to its canonical release in anaphase I, suggesting that mechanisms distinct from the standard anaphase-triggered release contribute to Cdc14 early function. This premature release of Cdc14 could be induced by DNA damage, as observed in mitotic cells [228], and may involve transient diffusion that previous studies may have missed due to technical limitations.

Further, the early release of Cdc14 may depend on post-translational modifications (PTMs) such as phosphorylation and SUMOylation. Phosphorylation by kinases such as Cdc28 and Cdc5, as well as the Mec1 checkpoint kinase, may be particularly relevant, as these modifications could influence early Cdc14 diffusion during the meiotic cell cycle. Further investigation is necessary to clarify these mechanisms and determine their impact on Cdc14 localization and meiotic recombination outcomes.

## 8. Concluding Remarks and Future Perspectives

Future studies will be crucial for deepening our understanding of how Cdc14 and other regulators contribute to meiotic recombination. Although much is known about the role of Cdc14 during anaphase I, further investigation into its behavior earlier in meiosis may uncover novel functions of the phosphatase regulated by the cell cycle. Emerging insights suggest that the nucleolus, as a biomolecular condensate, may influence the regulation of Cdc14 through mechanisms of phase separation, allowing for the selective sequestration and timed release of Cdc14. This control could be central to its function before anaphase I, potentially fine-tuning recombination. Exploring how phase separation within the nucleolus controls the availability and activity of Cdc14, particularly in response to recombination cues, could reveal uncharacterized layers of regulation within the cell cycle and meiosis. Additionally, a detailed analysis of interactions between Cdc14 and coregulators, including how phosphorylation and other modifications impact its release from the nucleolus, may provide a clearer picture of the signaling pathways that maintain genomic stability during gametogenesis. This research could advance our comprehension of the nucleolus as a regulatory hub in meiosis and clarify how phase separation contributes to precise enzyme control in reproductive cell division.

## Figures and Tables

**Figure 1 ijms-25-12861-f001:**
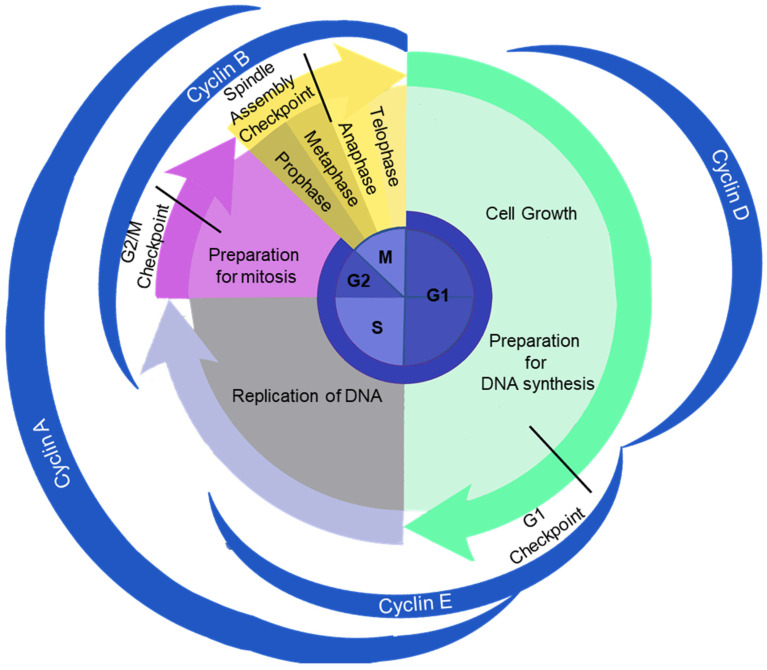
Cell cycle in eukaryotic cells. The four phases of the cell cycle, G_1_, S, G_2_, and M, and the main functions that take place in them are represented. The main checkpoints are also shown, as well as the main cyclins that act during the cycle.

**Figure 2 ijms-25-12861-f002:**
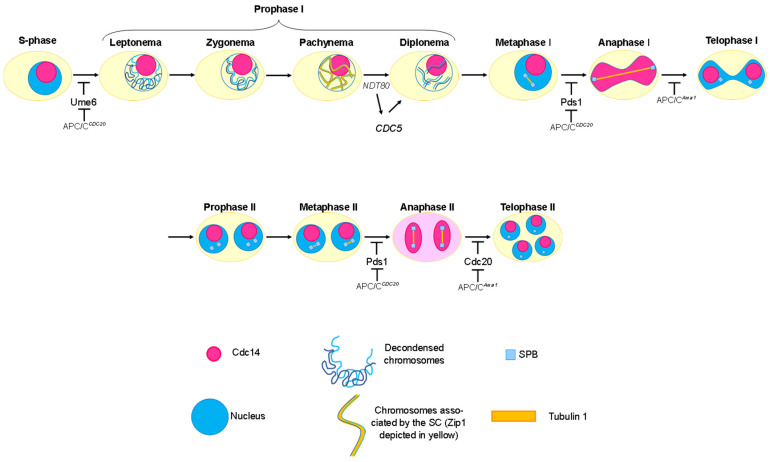
Diagram of the phases of meiosis and their regulation in *S. cerevisiae*. The nucleolus is represented in pink, the nucleus is in blue, and the SC is represented by its Zip1 component in yellow. The SPBs are represented by a blue square and the tubulin by a yellow rectangle. Arrows represent promotion of processes, while T-bars indicate prevention or inhibition.

**Figure 3 ijms-25-12861-f003:**
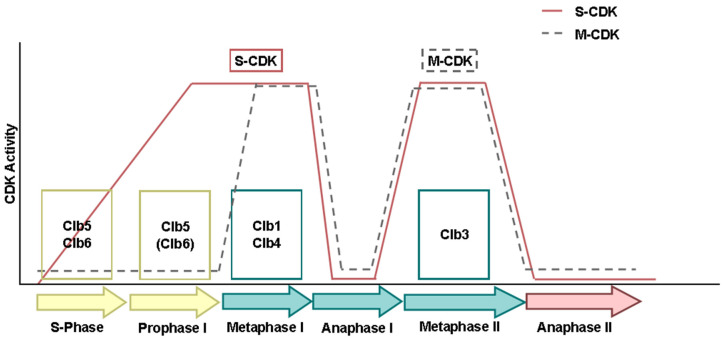
Variation in CDK activity levels of both S phase CDKs (S-CDKs) and M phase CDKs (M-CDKs). The arrows indicate the progression of meiosis, together with the cyclins that are acting at each point in time.

**Figure 4 ijms-25-12861-f004:**
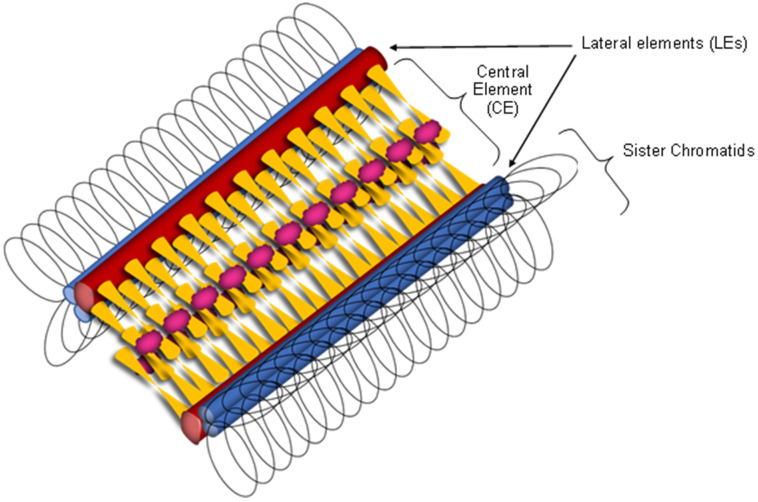
Diagram of the structure of the synaptonemal complex.

**Figure 5 ijms-25-12861-f005:**
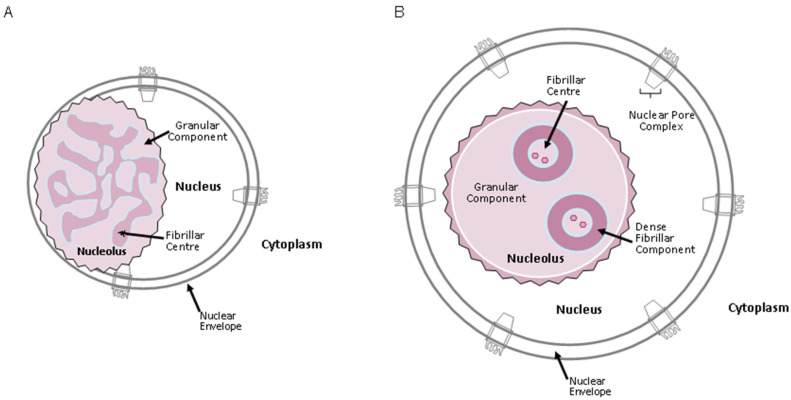
Models of nucleolus organization: (**A**) organization of the nucleolus of an *S. cerevisiae*; (**B**) organization of the nucleolus of a eukaryotic cell. The three subcompartments of the nucleolus are depicted: the fibrillar center, the dense fibrillar component, and the granular component.

**Figure 6 ijms-25-12861-f006:**
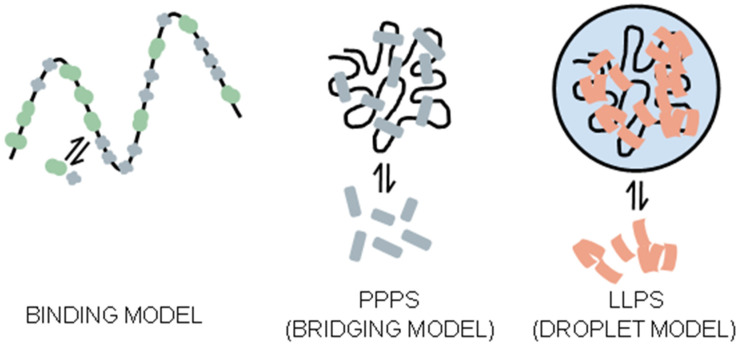
Schematic of the three models of organization of the membrane-less subcompartments. The nucleolus can be explained as a combination of the PPPS and LLPS model.

**Figure 7 ijms-25-12861-f007:**
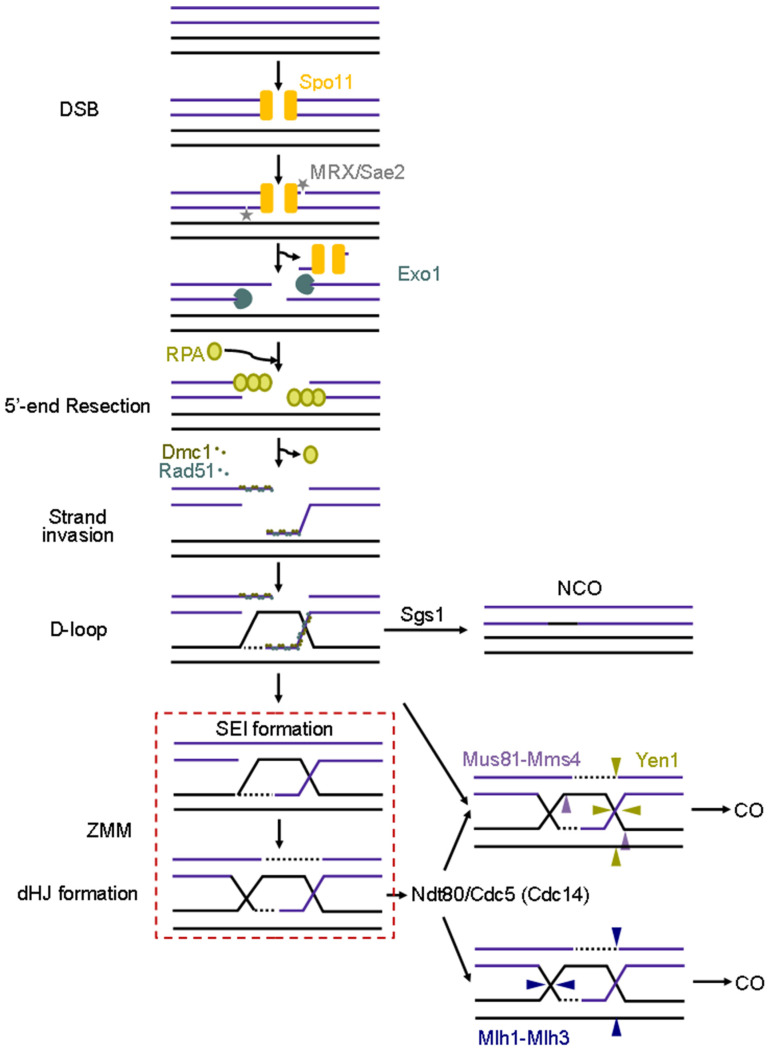
Schematic representation of the recombination process in *S. cerevisiae*.

**Figure 8 ijms-25-12861-f008:**
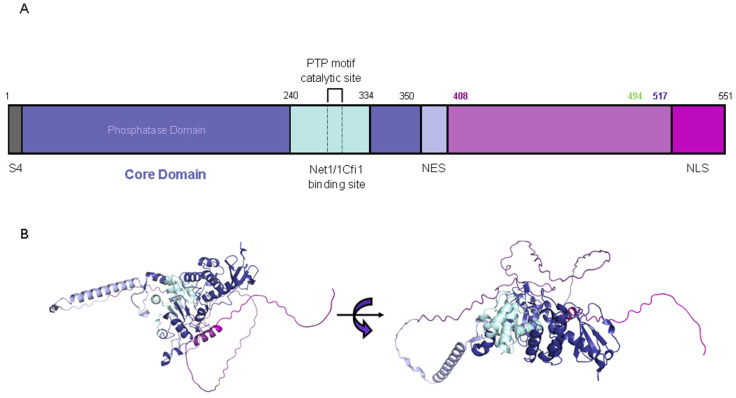
Domains and structure of Cdc14 phosphatase. (**A**) Schematic representation of the structure of Cdc14, with its main domains and binding sites. The phosphatase domain, the NES (nuclear export signal) and NLS (nuclear localization signal) region, and the 408,494,517 phosphorylation sites are shown. (**B**) Alphafold prediction model for Cdc14. Colors indicate the different functional motifs described in (**A**).

**Figure 9 ijms-25-12861-f009:**
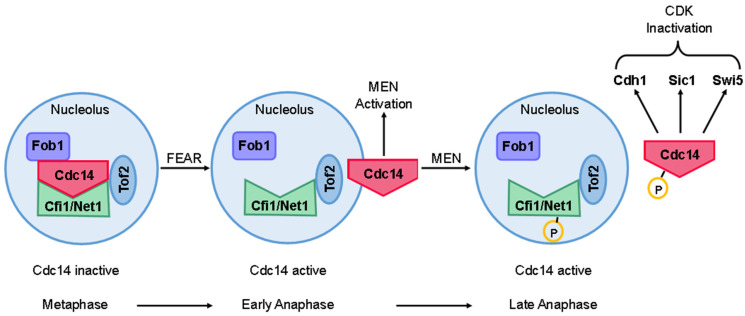
Role of Cdc14 in mitosis. The FEAR network initiates the early release of Cdc14, followed by the MEN pathway, which regulates its final release, enabling the dephosphorylation events crucial for completing mitosis. The large blue circle represents the nucleolus, with colors indicating key proteins involved in repressing Cdc14 release. The figure illustrates the two-wave pathway required for Cdc14 release, showing how the first wave also activates the MEN network. The right part displays how Cdc14, through Cdh1, Sic1, and Swi5, inactivates CDKs. The bottom of the figure shows how different mitotic phases correspond with the two waves of Cdc14 release.

**Figure 10 ijms-25-12861-f010:**
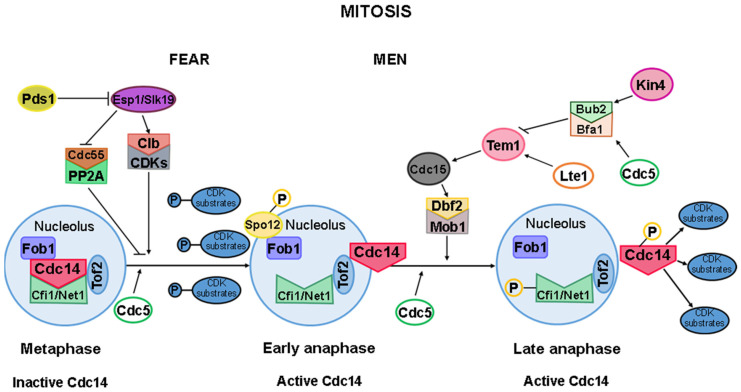
Schematic of Cdc14 regulation during mitosis via the FEAR and MEN pathways. The figure shows the early, partial release of Cdc14 from the nucleolus (blue circle) by the FEAR network, followed by its complete release through the MEN pathway. Proteins are depicted as circles, while proteins requiring a cofactor are represented as squares divided into two binding parts. The figure also illustrates CDK phosphorylation stages during Cdc14 release and the active or inactive states of Cdc14 throughout different mitotic phases.

**Figure 11 ijms-25-12861-f011:**
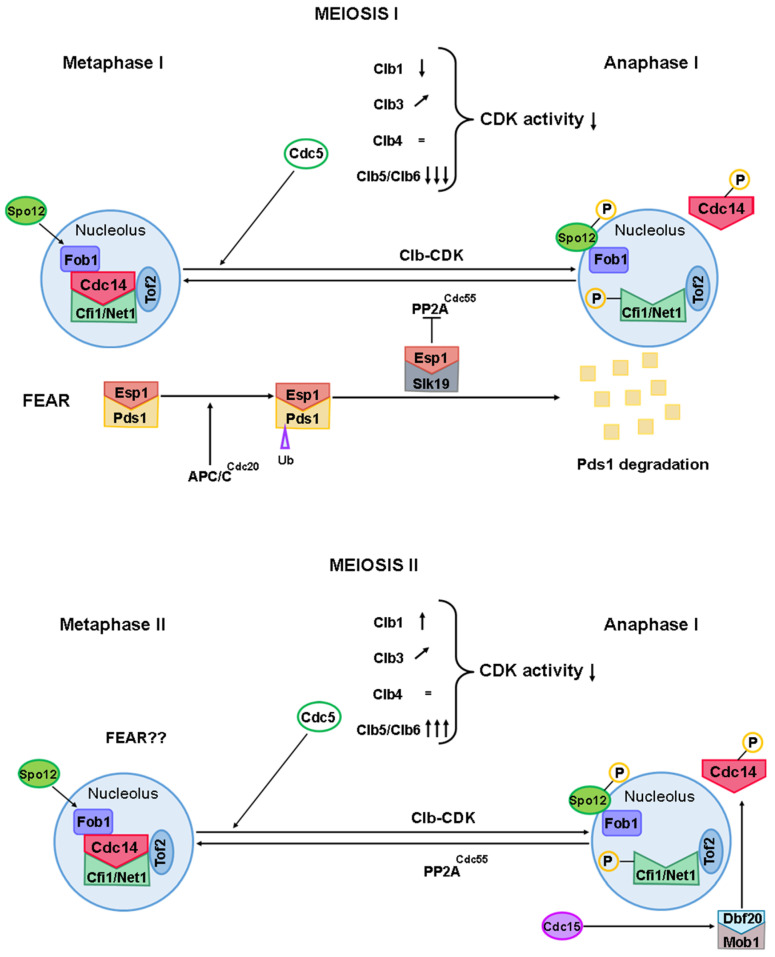
Schematic representation of Cdc14 release during the first and second meiotic divisions. The figure shows the key regulators and inhibitors required for Cdc14 release in meiosis I (top panel), along with variations in cyclin levels. Arrows indicate increases or decreases in cyclin levels, with multiple arrows representing a stronger effect. An oblique arrow pointing up denotes a progressive increase in cyclin levels, while the “=” symbol indicates no change. These adjustments lead to a downregulation of CDK activity. In meiosis II (bottom panel), the role of the FEAR pathway in Cdc14 release is uncertain (noted as FEAR??). Cyclin level changes are similarly depicted, illustrating how they contribute to CDK activity downregulation.

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
