# Peer review of "Decoding the Nucleolar Role in Meiotic Recombination and Cell Cycle Control: Insights into Cdc14 Function"

_ijms, 2024, doi:10.3390/ijms252312861_

Round 1

Reviewer 1 Report

Comments and Suggestions for Authors

« Decoding the Nucleolar Role in Meiotic Recombination and Cell Cycle Control: Insights into Cdc14 Function » by Paula Alonso-Ramos & Jesús A. Carballo

In this manuscript, the authors begin with a brief description of mitosis, followed by a lengthy description of meiosis with some comparisons. Throughout the review, they focus on the yeast S cerevisiae. The review also describes the nucleolus, how this membrane-less structure is formed and what its function is. Finally, the authors focus on Cdc14, a phosphatase sequestered in the nucleolus and released in anaphase I and anaphase II to relocate to SBP, asymmetrically in Ana-I and symmetrically in Ana-II.

I have no comments to make, except in the introduction with three suggestions.

1 - 

The cell cycle is organized into four distinct phases: interphase—comprising the G1 phase, where the cell grows and prepares for DNA synthesis; the S phase, during which DNA replication occurs; and the G2 phase, where the cell readies itself for mitosis—and mitosis, where replicated DNA is accurately segregated and transmitted to progeny cells (Figure 1; [1]).  

The second stage of the cell cycle, mitosis, is further divided into several phases based 49 on the level of chromosomal condensation and the status of the nuclear envelope. Mitosis 50 is subdivided into: 

- Prophase, characterized by the condensation and individualization of chromosomes. 

- Metaphase, during which chromosomes attain their maximum level of compaction. 

- Anaphase, where chromosomes are equally segregated to opposite poles of the cell. 

- Telophase, following segregation, chromosomes decondense, and the process of cell division, or cytokinesis, ensues

My suggestion

The cell cycle is organised into four distinct phases: G1, S, G2 and M. During the G1 phase, the cell develops and prepares for DNA synthesis, which takes place during the S phase. During the G2 phase, the cell prepares for mitosis, during which replicated DNA is precisely segregated and passed on to descendant cells. (Figure 1; [1]).

Mitosis is subdivided into 5 sub-phases :  

- Prophase, characterized by the condensation and individualization of chromosomes and by nuclear envelope breakdown. 

- Prometaphase identified by bipolar spindle assembly and progressive alignment of chromosomes

- Metaphase, which is not really a phase since it corresponds to the achievement of chromosome alignment that immediately triggers anaphase. 

- Anaphase A and B, where chromosomes are equally segregated to opposite poles of the cell. 

- Telophase, during chromosomes decondense and the cell prepare cell division, or cytokinesis.

2 - 

The cell cycle is a tightly regulated process that relies on molecular mechanisms promoting cell cycle progression, along with checkpoints that act as negative regulators. These checkpoints halt cell cycle progression if the preceding phase is incomplete or if errors are detected.

My suggestion

The cell cycle is a tightly regulated process that relies on molecular mechanisms promoting cell cycle progression, along with checkpoints that act as quality controls. These checkpoints slow down or halt cell cycle progression if the preceding phase is incomplete or if errors are detected.

3 -

The cell cycle can be studied using various model organisms, such as Saccharomyces cerevisiaeSchizosaccharomyces pombeXenopus laevis, and Drosophila melanogaster.

As the cell cycle and its controls are universally conserved in eukaryotes, they have been studied using various model organisms, starting with the yeasts Saccharomyces cerevisiaeSchizosaccharomyces pombe, and amphibians such as Xenopus laevis, and the fly Drosophila melanogaster.

Author Response

Response to Reviewer 1 comments:

I have no comments to make, except in the introduction with three suggestions.

COMMENT 1 - 

The cell cycle is organized into four distinct phases: interphase—comprising the G1 phase, where the cell grows and prepares for DNA synthesis; the S phase, during which DNA replication occurs; and the G2 phase, where the cell readies itself for mitosis—and mitosis, where replicated DNA is accurately segregated and transmitted to progeny cells (Figure 1; [1]).  

The second stage of the cell cycle, mitosis, is further divided into several phases based 49 on the level of chromosomal condensation and the status of the nuclear envelope. Mitosis 50 is subdivided into: 

- Prophase, characterized by the condensation and individualization of chromosomes. 

- Metaphase, during which chromosomes attain their maximum level of compaction. 

- Anaphase, where chromosomes are equally segregated to opposite poles of the cell. 

- Telophase, following segregation, chromosomes decondense, and the process of cell division, or cytokinesis, ensues

My suggestion

The cell cycle is organised into four distinct phases: G1, S, G2 and M. During the G1 phase, the cell develops and prepares for DNA synthesis, which takes place during the S phase. During the G2 phase, the cell prepares for mitosis, during which replicated DNA is precisely segregated and passed on to descendant cells. (Figure 1; [1]).

Mitosis is subdivided into 5 sub-phases :  

- Prophase, characterized by the condensation and individualization of chromosomes and by nuclear envelope breakdown. 

- Prometaphase identified by bipolar spindle assembly and progressive alignment of chromosomes

- Metaphase, which is not really a phase since it corresponds to the achievement of chromosome alignment that immediately triggers anaphase. 

- Anaphase A and B, where chromosomes are equally segregated to opposite poles of the cell. 

- Telophase, during chromosomes decondense and the cell prepare cell division, or cytokinesis.

RESPONSE 1: We appreciate the reviewer’s suggestion, which has been very helpful in improving this manuscript. We have incorporated most of the recommended changes, making slight adjustments to a few definitions to better align with our understanding of certain phases. These modifications aim to ensure clarity and consistency throughout the text.

COMMENT 2 - 

The cell cycle is a tightly regulated process that relies on molecular mechanisms promoting cell cycle progression, along with checkpoints that act as negative regulators. These checkpoints halt cell cycle progression if the preceding phase is incomplete or if errors are detected.

My suggestion

The cell cycle is a tightly regulated process that relies on molecular mechanisms promoting cell cycle progression, along with checkpoints that act as quality controls. These checkpoints slow down or halt cell cycle progression if the preceding phase is incomplete or if errors are detected.

RESPONSE 2: We have now incorporated the recommended changes in the revised text.

COMMENT 3 -

The cell cycle can be studied using various model organisms, such as Saccharomyces cerevisiaeSchizosaccharomyces pombeXenopus laevis, and Drosophila melanogaster.

As the cell cycle and its controls are universally conserved in eukaryotes, they have been studied using various model organisms, starting with the yeasts Saccharomyces cerevisiaeSchizosaccharomyces pombe, and amphibians such as Xenopus laevis, and the fly Drosophila melanogaster.

RESPONSE 3: We have now incorporated the recommended changes in the revised text.

Reviewer 2 Report

Comments and Suggestions for Authors

Review on the manuscript of Alonso-Ramos P et al., (ijms-3306142): “Decoding the Nucleolar Role in Meiotic Recombination and Cell Cycle Control: Insights into Cdc14 Function”.

In this manuscript, the Authors review the current knowledge on the mechanisms that govern cell cycle regulation and meiotic recombination.

Overall, I find this topic highly engaging, as exploring the nucleolar role in meiotic recombination and cell cycle control with a focus on Cdc14 function is crucial for understanding key aspects of cellular division, genome stability, and organismal health. This manuscript provides a comprehensive review on an important topic, which would be valuable reading for Students and Researchers in this field. The Authors have effectively addressed the main question posed, and the manuscript is well-written and well-organized. My suggestions for improvement are listed below, which I hope the Authors find helpful.

1 - The main goal of the manuscript is to provide an overview of the nucleolar role in meiotic recombination and cell cycle control. However, in some sections, the Authors go into detail on mitosis (e.g., in the section “Role of Cdc14 in mitosis”). It might be more effective to keep the focus on meiosis, and rather than discussing mitosis, further explore the relevance of these mechanisms in other organisms. While the Authors already make connections to other organisms in parts of the manuscript, this approach could be applied more consistently throughout.

2 - It seems that the colors in panel 8B don’t fully align with those in panel 8A. For example, the yellow domain in panel A isn’t represented in B, and similarly, the green domain in panel B doesn’t appear in A.

3 - For Figures 9 and 10, I believe it would be beneficial for the readers to include a description in the figure legend explaining what is represented. In these figures, several elements are indicated, and without a clear description, interpreting the figures could be more challenging.

4 - I believe that a more detailed discussion on how Cdc14’s regulation of recombination affects chromosomal integrity and genome stability would greatly enhance the manuscript. Although this is somewhat implicit in the current text, the Authors could place additional emphasis on it, perhaps by creating a dedicated section to better highlight this critical aspect.

5 - It might be helpful to consider adding a concluding section to the manuscript, as it could enhance overall coherence and provide a clear summary of the key insights.

6 - It might be helpful to consider adding a future perspectives section to the manuscript. For instance, this section could provide unanswered questions and gaps in understanding Cdc14’s nucleolar functions and cross-talk with other cell cycle regulators.

Minor points:

1 - It looks like the abstract might exceed the limit of 200 words required by the journal.

2 - It seems that “Double Strand Breaks (DSBs)” is mentioned twice, on line 108 (page 3) and line 93 (page 8). Would the Authors be able to revise this?

3 - On line 231 (page 11), “ZYP1” is written in capital letters, while on line 233 it is not. Could the Authors standardize this?

4 - In Figure 11, can the authors indicate what the oblique arrow represents?

Author Response

Response to Reviewer 2 comments:

In this manuscript, the Authors review the current knowledge on the mechanisms that govern cell cycle regulation and meiotic recombination.

Overall, I find this topic highly engaging, as exploring the nucleolar role in meiotic recombination and cell cycle control with a focus on Cdc14 function is crucial for understanding key aspects of cellular division, genome stability, and organismal health. This manuscript provides a comprehensive review on an important topic, which would be valuable reading for Students and Researchers in this field. The Authors have effectively addressed the main question posed, and the manuscript is well-written and well-organized.

GENERAL RESPONSE: Thank you for your positive feedback and for highlighting the relevance and value of our manuscript. We are glad you found the review engaging and well-organized, and we appreciate your recognition of its potential as a resource for students and researchers in the field.

My suggestions for improvement are listed below, which I hope the Authors find helpful.

COMMENT 1 - The main goal of the manuscript is to provide an overview of the nucleolar role in meiotic recombination and cell cycle control. However, in some sections, the Authors go into detail on mitosis (e.g., in the section “Role of Cdc14 in mitosis”). It might be more effective to keep the focus on meiosis, and rather than discussing mitosis, further explore the relevance of these mechanisms in other organisms. While the Authors already make connections to other organisms in parts of the manuscript, this approach could be applied more consistently throughout.

RESPONSE 1: We appreciate the reviewer’s concern and fully agree that meiosis is the primary focus of our study. However, the majority of the well-established knowledge regarding Cdc14 has been derived from research on mitotic cells, with relatively little parallel information available for meiosis. As such, we believe it is important to first highlight the key findings from mitosis to provide a solid foundation for understanding Cdc14 function. We then explore whether these roles are conserved in meiosis, offering specific examples where they appear to be conserved or differ, to provide a more complete picture.

COMMENT 2 - It seems that the colors in panel 8B don’t fully align with those in panel 8A. For example, the yellow domain in panel A isn’t represented in B, and similarly, the green domain in panel B doesn’t appear in A.

RESPONSE 2: Thank you for your helpful feedback. We understand that the colour output from the ChimeraX palette (used for the Alphafold2 model of Cdc14) has some limitations compared to the flexibility of our graphic design tool. To improve this, we have adjusted the colours in Figure 8 to make them consistent across panels A and B. We also removed unnecessary features that could have caused confusion. These changes make the revised Figure 8 clearer and easier to understand.

COMMENT 3 - For Figures 9 and 10, I believe it would be beneficial for the readers to include a description in the figure legend explaining what is represented. In these figures, several elements are indicated, and without a clear description, interpreting the figures could be more challenging.

RESPONSE 3: We have now included more detailed revised figure legends for figures 9 and 10.

COMMENT 4 - I believe that a more detailed discussion on how Cdc14’s regulation of recombination affects chromosomal integrity and genome stability would greatly enhance the manuscript. Although this is somewhat implicit in the current text, the Authors could place additional emphasis on it, perhaps by creating a dedicated section to better highlight this critical aspect.

RESPONSE 4. We have now renamed the last section of the manuscript from “Alternative roles of Cdc14 and cell-cycle regulators in meiosis” by the “Roles of Cdc14 in meiotic recombination”, and we have expanded the content on such topic, including adding a new reference.

COMMENT 5 - It might be helpful to consider adding a concluding section to the manuscript, as it could enhance overall coherence and provide a clear summary of the key insights.

RESPONSE 5: see response 6.

COMMENT 6 - It might be helpful to consider adding a future perspectives section to the manuscript. For instance, this section could provide unanswered questions and gaps in understanding Cdc14’s nucleolar functions and cross-talk with other cell cycle regulators.

RESPONSE 6: We decided to combine the concluding remarks and future perspectives into one last section to keep the review concise and avoid adding too many extra sections. In this section, we included key insights from the manuscript while also addressing some unanswered questions about Cdc14’s nucleolar functions and its interactions with other cell cycle regulators. We felt this would help highlight the critical aspects of the review without expanding too much on the overall structure.

Minor points:

Minor comment 1 - It looks like the abstract might exceed the limit of 200 words required by the journal.

Response to minor comment 1: We have now reduced the length of the abstract to a more reasonable word count.

Minor comment 2 - It seems that “Double Strand Breaks (DSBs)” is mentioned twice, on line 108 (page 3) and line 93 (page 8). Would the Authors be able to revise this?

Response to minor comment 2: Thank you for pointing out the duplication. We have now removed the second instance and kept only the acronym.

Minor comment 3 - On line 231 (page 11), “ZYP1” is written in capital letters, while on line 233 it is not. Could the Authors standardize this?

Response to minor comment 3: Thank you for your comment, and apologies for any confusion. We used organism-specific nomenclature for the gene/protein names—ZYP1 refers to the plant protein, while Zip1, in lane 233, refers to the yeast protein. We hope this clarifies the distinction. However, if necessary, we could use AtZYP1 and ScZip1 to specify each version, although we feel this might complicate the list of proteins in different organisms, as protein and gene names sometimes use even contradictory nomenclatures from one organism to another. We believe the current naming convention is the clearest option.

Minor comment 4 - In Figure 11, can the authors indicate what the oblique arrow represents?

Response to minor comment 4: We apologize for this oversight. The oblique arrows following the Clb3 cyclin name indicate a progressive increase of cyclin levels through meiosis. This clarification has now been added to the revised Figure legend.

Round 2

Reviewer 2 Report

Comments and Suggestions for Authors

Second review on the manuscript of Alonso-Ramos P et al., (ijms-3306142): “Decoding the Nucleolar Role in Meiotic Recombination and Cell Cycle Control: Insights into Cdc14 Function”.

In this manuscript, the Authors review the current knowledge on the mechanisms that govern cell cycle regulation and meiotic recombination.

This represents a second version of the manuscript after peer review. The Authors have effectively made satisfactory modifications to the manuscript, in line with the Reviewers' comments. There is just a minor suggestion provided below that would help clarify the content of the manuscript.

1 - I agree that the majority of the well-established knowledge regarding Cdc14 has been derived from research on mitotic cells, with relatively little parallel information available for meiosis. Thus, it would be more appropriate to change the title to reflect the content of the manuscript. As alternative would be “decoding the nucleolar role in meiotic recombination and cell cycle control: Insights into Cdc14 function and from mitosis”.

Author Response

COMMENT 1: I agree that the majority of the well-established knowledge regarding Cdc14 has been derived from research on mitotic cells, with relatively little parallel information available for meiosis. Thus, it would be more appropriate to change the title to reflect the content of the manuscript. As alternative would be “decoding the nucleolar role in meiotic recombination and cell cycle control: Insights into Cdc14 function and from mitosis”.

RESPONSE 1: 

Thank you for your thoughtful feedback and suggestion to change the title. This study focuses on the less understood role of Cdc14 in meiosis, while also building on the well-established knowledge from studies in mitotic cells, as already discussed.

We believe the current title, "Decoding the nucleolar role in meiotic recombination and cell cycle control: Insights into Cdc14 function," already explains this focus well. The phrase "cell cycle" already includes the idea of mitosis as part of cell proliferation. Adding "mitosis" to the title would make it redundant and take attention away from the less-studied meiotic role of Cdc14, which is the main focus of this review.

For these reasons, we would prefer to keep the current title, which we believe clearly reflects the manuscript's focus. Thank you again for your helpful suggestions and the opportunity to explain this further.